# Quantitative Diffusion-Weighted MRI of Neuroblastoma

**DOI:** 10.3390/cancers15071940

**Published:** 2023-03-23

**Authors:** Niklas Abele, Soenke Langner, Ute Felbor, Holger Lode, Norbert Hosten

**Affiliations:** 1Department of Radiology, Germany University of Greifswald, 17475 Greifswald, Germany; 2Institute of Pathology, University of Erlangen, 91054 Erlangen, Germany; 3Department of Radiology, University of Rostock, 18057 Rostock, Germany; 4Department of Human Genetics, University of Greifswald, 17475 Greifswald, Germany; 5Interfaculty Institute of Genetics and Functional Genetics, University of Greifswald, 17475 Greifswald, Germany; 6Department of Pediatric Hematology and Oncology, University of Greifswald, 17475 Greifswald, Germany

**Keywords:** neuroblastoma, DWI, MRI, biomarker, quantitative image analysis, pathology

## Abstract

**Simple Summary:**

In this study quantified diffusion weitgthed imaging in MRI was analized in neuroblastoma. We were able to show a significant increase of apparent diffusion coefficient in regressive diseases and a decrease for progressive diseases. This was even true within the first 120 days after the start of therapy.

**Abstract:**

Neuroblastoma is the most common extracranial, malignant, solid tumor found in children. In more than one-third of cases, the tumor is in an advanced stage, with limited resectability. The treatment options include resection, with or without (neo-/) adjuvant therapy, and conservative therapy, the latter even with curative intent. Contrast-enhanced MRI is used for staging and therapy monitoring. Diffusion-weighted imaging (DWI) is often included. DWI allows for a calculation of the apparent diffusion coefficient (ADC) for quantitative assessment. Histological tumor characteristics can be derived from ADC maps. Monitoring the response to treatment is possible using ADC maps, with an increase in ADC values in cases of a response to therapy. Changes in the ADC value precede volume reduction. The usual criteria for determining the response to therapy can therefore be supplemented by ADC values. While these changes have been observed in neuroblastoma, early changes in the ADC value in response to therapy are less well described. In this study, we evaluated whether there is an early change in the ADC values in neuroblastoma under therapy; if this change depends on the form of therapy; and whether this change may serve as a prognostic marker. We retrospectively evaluated neuroblastoma cases treated in our institution between June 2007 and August 2014. The examinations were grouped as ‘prestaging’; ‘intermediate staging’; ‘final staging’; and ‘follow-up’. A classification of “progress”, “stable disease”, or “regress” was made. For the determination of ADC values, regions of interest were drawn along the borders of all tumor manifestations. To calculate ADC changes (*∆ADC),* the respective MRI of the prestaging was used as a reference point or, in the case of therapies that took place directly after previous therapies, the associated previous staging. In the follow-up examinations, the previous examination was used as a reference point. The *∆ADC* were grouped into *∆ADCregress* for regressive disease, *∆ADCstable* for stable disease, and *∆ADC* for progressive disease. In addition, examinations at 60 to 120 days from the baseline were grouped as *er∆ADCregress*, *er∆ADCstable*, and *er∆ADCprogress*. Any differences were tested for significance using the Mann–Whitney test (level of significance: *p* < 0.05). In total, 34 patients with 40 evaluable tumor manifestations and 121 diffusion-weighted MRI examinations were finally included. Twenty-seven patients had INSS stage IV neuroblastoma, and seven had INSS stage III neuroblastoma. A positive N-Myc expression was found in 11 tumor diseases, and 17 patients tested negative for N-Myc (with six cases having no information). 26 patients were assigned to the high-risk group according to INRG and eight patients to the intermediate-risk group. There was a significant difference in mean ADC values from the high-risk group compared to those from the intermediate-risk group, according to INRG. The differences between the mean *∆ADC* values (absolute and percentage) according to the course of the disease were significant: between *∆ADCregress* and *∆ADCstable*, between *∆ADCprogress* and *∆ADCstable*, as well as between *∆ADCregress* and *∆ADCprogress*. The differences between the mean *er∆ADC* values (absolute and percentage) according to the course of the disease were significant: between *er∆ADCregress* and *er∆ADCstable*, as well as between *er∆ADCregress* and *er∆ADCprogress*. Forms of therapy, N-Myc status, and risk groups showed no further significant differences in mean ADC values and *∆ADC*/*er∆ADC*. A clear connection between the ADC changes and the response to therapy could be demonstrated. This held true even within the first 120 days after the start of therapy: an increase in the ADC value corresponds to a probable response to therapy, while a decrease predicts progression. Minimal or no changes were seen in cases of stable disease.

## 1. Introduction

Neuroblastoma, an embryonic tumor originating in the sympathetic nervous system, is the most common extracranial malignant solid tumor in children [1]. Neuroblastoma is said to have a high degree of variability in terms of site of manifestation and prognosis. In about 37% of cases, the tumor is already at an advanced stage with limited resectability or metastatic spread when it is first diagnosed. The overall ten-year survival rate in a 2004 cohort was 72% [1].

Primary manifestations are mostly found in children younger than six years old and are mainly localized in the adrenal glands and the sympathetic trunk or the paraganglia [2]. The histopathological examination using an open biopsy offers the highest level of certainty for confirming the diagnosis and is considered the procedure of choice [2,3]. Histologically, the tumor is predominantly solid and made up of densely packed, round-celled, basophilic cells, with several subtypes being recognized that can be classified according to International Neuroblastoma Pathology Classification criteria [2,3,4]. Among other indicators, a lower proportion of stroma correlates with a more aggressive course of the disease [3,4]. As part of the tissue examination, an analysis for N-Myc amplification and possible chromosome aberrations is usually also carried out [5]. In particular, tumors with N-Myc amplification are said to have a poorer prognosis [6,7,8,9].

In the initial tumor staging of the neuroblastoma, whole-body or magnetic resonance imaging (MRI) of the affected body regions is planned.

An essential part of the MRI staging examination should include staging according to the INSS (International Neuroblastoma Staging System) and risk stratification according to the INRGSS (International Neuroblastoma Risk Group Staging System) [10,11,12]. Other diagnostic standards or supplementary examinations include ultrasound examinations, X-ray examinations including computed tomography, Tc^99m^-scintigraphy, ^123^I- and ^131^I-MIBG, and ^18^F-flurodeoxyglucose PET/CT [13].

Neuroblastoma diseases are usually treated within the framework of studies and are based on the respective study protocols. Depending on the stage and IDRFs, the therapy regimen usually includes tumor resection. In many cases, further adjuvant therapy follows. In the absence of resectability, non-surgical forms of therapy can be used primarily and with a curative approach [14]. In addition to chemotherapy (including high-dose therapy cycles with autologous stem cell transplantation) and radiotherapy, ^131^I-MIBG therapy and immunological therapy using antibody administration (AB) are used [2,15,16].

Even if a uniform MRI protocol in childhood tumor diseases has not yet been established, T2-weighted (T2w) and T2w fat-saturated (T2w fs) images (e.g., STIR), as well as T1-weighted (T1w) images, acquired before and after the administration of a gadolinium-based contrast agent, are standard practice. In addition, diffusion-weighted sequences are utilized [17]. Diffusion-weighted imaging (DWI) allows the visualization of functional tissue properties, providing information about the diffusion of water in the tissue in vivo [18,19]. If images with at least two *b*-values are recorded in a diffusion-weighted sequence, the so-called ADC value (apparent diffusion coefficient) can be calculated in the form of a parameter map. In the calculated ADC parameter maps, facilitated diffusion, e.g., in the context of edema due to inflammation, appears as an increased ADC value and is therefore light, while restricted diffusion with a reduced ADC value, e.g., in cell-rich tumors or a stroke, appears dark [20].

A connection between diffusion or the ADC value and the histological properties of tissues, such as the cell density of tumors, has been shown several times [21]. For example, it has been shown that carcinomas of the ENT area or tumors of the breast could be classified and evaluated more accurately with the help of diffusion-weighted imaging [22,23,24]. The positive correlation between the tumor stroma content and ADC value and the negative correlation between tumor cellularity and the ADC value should be emphasized [22].

DWI also comes into play when monitoring the response to the treatment of malignant lesions; among other observations, a significant increase in the ADC value was found when hepatocellular carcinoma responded to chemotherapy as an expression of the altered ultrastructure. The changes in the ADC value were already measurable before a reduction in the size of the tumor could be observed. The usual criteria for determining the response to therapy can therefore be supplemented by considering ADC values in order to possibly make an earlier statement regarding the response to therapy [25].

If one considers previous findings on the importance of DWI in neuroblastoma, a less well-studied but comparable picture emerges. In 2014, Demir et al. showed that the ADC value of eleven examined neuroblastomas under chemotherapy increased significantly in response to therapy [26]. However, no statement could be made about an early change in the ADC value that could be measured before the end of the approximately five-month therapy to indicate the response to therapy [27]. In 2017, Neubauer et al., taking into account similar previous studies, found that the ADC value of neuroblastoma differed from the ADC values of neuroblastic tumors with lower malignancy in childhood that were eligible for differential diagnosis. The DWI can therefore be of great help in the diagnosis of such lesions [28].

The current study answers the following questions:Is there an (early) change in the ADC values in neuroblastoma under therapy as an expression of therapy-associated changes in the ultrastructure of the tumor?If there is a change in the ADC values of the tumors, does this depend on the form of therapy?Can therapy-associated changes in the ADC value serve as a prognostic marker in patients with neuroblastoma?

## 2. Materials and Methods

As part of the study, all in-house and external MRI examinations of patients with neuroblastoma disease from the period June 2007 to August 2014 were retrospectively evaluated. For this purpose, the collective was taken from the register of the Clinic for Pediatric and Adolescent Medicine, Greifswald.

The inclusion criteria were the following:The presence of a histologically confirmed neuroblastoma disease with at least one solid manifestation and at least two MRI examinations in the specified period so that the tumor manifestations to be examined were presented at least twice.If an MIBG scintigraphy was available, a tracer recording of the tumor manifestation was required that could be delineated in the MRI in the MIBG scintigraphy (MIBG).Available MRI data, including pre- and post-contrast T1w sequences, T2w sequences, and DWI images, were required.

The clinical parameters were taken from interdisciplinary tumor conferences and from external doctors’ letters.

Each patient was placed into a risk group according to the INRG; the stage according to INSS was also taken from the protocols of the tumor boards or the doctors’ letters if the relevant information was available. In the absence of information (four cases), the classification for the study was defined post hoc by a radiological specialist (>10 years of professional experience). The IDRFs were also taken from the protocols of the tumor boards and the doctors’ letters, as well as the radiological findings. In the absence of sufficient information regarding IDRFs, no IDRFs were reported. If possible, the INRG stage was taken directly from the protocols of the tumor boards or the doctors’ letters. Otherwise, it was determined post hoc from the IDRFs. If neither the INRG stage nor the IDRFs were known, the INRG stage was determined according to INRG Cohn et al. [11] from the INSS stage. Finally, the patients were classified into risk groups (high, intermediate, low, and very low) according to INRG Cohn et al. In the case of insufficient information and missing factors, the value was noted, which resulted in a higher classification when dividing the patients into risk groups. In addition, the forms of therapy carried out were recorded in detail.

Furthermore, if available, the molecular genetic N-Myc status (basic helix–loop–helix protein 37) was also recorded.

Based on the data collected in this way, the MRI examinations were divided into the following points in time:Prestaging: examination before the start of the respective form of therapy.Intermediate staging: examination during ongoing therapy.Final staging: examination after completion of the respective form of therapy.Follow-up: follow-up examination after the final staging without any therapy reference.Other: investigation without known cause for investigation.

To assess the course of the disease, a classification was made, i.e., “progressive disease” (PD) and “stable disease” (StD) or “partial response” (PR) and “complete response” (CR), according to the INRC based on the respective final staging, whereby the categories CR and PR were combined (regress) [29,30]. The categorization was based on the extent of the tumor or the absence and occurrence of possible new manifestations in the MRI and was also compared with the written assessment of the respective examination if this was available. As a reference point or baseline (BL), the respective prestaging, or, in the case of therapies that took place directly after previous therapies, the associated previous staging was used. For follow-up examinations, the previous examination was used as a reference point.

If there was no final staging examination that met the inclusion criteria mentioned, any findings from the imaging examinations carried out were used solely for the purpose of assessing the course of the disease.

In the event of disagreement between the radiological or nuclear medicine findings and the clinical assessment, the consensus decision of the interdisciplinary tumor conferences was used to assess the response to therapy.

Since the MRI examinations were carried out both in-house and externally, the examination protocols and sequence parameters used varied accordingly. However, all investigations were performed at 1.5 T using a surface coil for signal detection.

Diffusion-weighted images with the diffusion factors b = 0/800 s/mm^2^ were available for all included patients. If the calculated ADC parameter map was not available, it was calculated using the ADC plug-in (https://github.com/mribri999/ADCmap) (accessed on 15 October 2014) in OsiriX (v.3.4.1, Pixmeo SARL, Bernex, Switzerland).

The collected image data of the patient collective were transferred to an OsiriX workstation (v.3.4.1, Pixmeo SARL, Bernex, Switzerland) for further evaluation.

To determine the ADC values, a “closed polygon”, a region of interest (ROI) along the borders of the tumor manifestation, was drawn on the axial slice images. The tumor manifestation was identified according to the respective radiological findings or the tumor board protocol. If multiple tumor manifestations were present, the largest solid manifestation was selected for each organ. An analogous procedure was chosen for lymph node manifestations. For the definition of the ROI, the anatomical sequence in which the tumor could best be delineated was chosen. This selection was the responsibility of the evaluator.

The ADC parameter map was then fused with the image of the selected anatomical sequence using the OsiriX fusion tool (v.3.4.1, Pixmeo SARL, Bernex, Switzerland), and the ROI was, thus, copied onto the ADC parameter map. Attention was paid to the anatomical congruence of both sequences. If there was a mismatch, the ROI was corrected manually.

In addition, the tumor volume was determined for each observed tumor manifestation. For this purpose, in the axial sequence with the best visibility of the respective tumor manifestation, the tumor was manually drawn with the “closed polygon” drawing tool in OsiriX (see Figure 1), and then the “compute volume” function was used to calculate the volume semi-automatically.

The continuous variables are presented as the mean ± standard deviation.

For further evaluation, three categories were formed from the determined ADC values and the calculated differences according to the course of the disease:

*∆ADCregress*: the difference between the ADC value of the examined manifestation determined by means of ROI analysis and the associated ADC value of the corresponding reference point at regress.

*∆ADCstable*: the difference between the ADC value of the examined manifestation determined by means of ROI analysis and the associated ADC value of the corresponding reference point at stable disease.

*∆ADCprogress*: the difference between the ADC value of the examined manifestation determined by means of ROI analysis and the associated ADC value of the corresponding reference point at progress.

Additional *∆ADC* values were selected whose calculation involved the use of an examination during therapy, providing this examination took place at least 60 and at most 120 days from the BL of the respective therapy. (Now called *er∆ADCregress*, *er∆ADCstable*, and *er∆ADCprogress,* respectively.)

These values were also compared with one another, considering the clinical parameters (N-Myc, age, gender, type of therapy, stage, tumor size, tumor location, primary/secondary/lymph node metastasis/recurrence) and the absolute ADC values.

For illustration, the results are presented using box plots, histograms, and ROCs.

Differences were tested for significance using the Mann–Whitney test. A significance level of *p* < 0.05 was set a priori.

The statistical evaluation was carried out using Microsoft Excel (v.1808, Microsoft Office Professional 2019, Redmond, WA, USA) and SPSS Statistics (v.24, SPSS IBM, Armonk, NY, USA).

The responsible ethics committee in Greifswald approved the study. Due to the retrospective nature of the study, written consent was not required.

## 3. Results

The neuroblastoma registry of the Clinic for Pediatrics and Adolescent Medicine at the University Medical Center, Greifswald, included 113 patients. Imaging was not available for 15 patients. Of the 98 patients with available imaging, 45 patients were excluded because no diffusion-weighted images were available. A further 13 of the remaining 53 patients were excluded because the diffusion-weighted images were only available from one study time point. A further six patients were excluded because the tumor-suspected lesions could not be evaluated with certainty due to their size in the diffusion-weighted images or because the lesions in the MIBG scintigraphy were negative; thus, 34 patients remained available for the final analysis (see Figure 2).

The youngest patient was 12 months old at the time of the first imaging, and the oldest patient was 26 years old. (Average age at the time of the initial examination 6.8 ± 5.8 a.)

Twenty-seven patients had INSS stage IV neuroblastoma, and seven had INSS stage III neuroblastoma.

Two or more IDRFs were present in 22 patients, and ≤1 IDRF was present in four. In eight cases, there was no information regarding IDRFs. Positive N-Myc expression was present in 11 tumor diseases, and 17 patients tested negative for N-Myc. In six cases, there was no information on N-Myc status.

Twenty-six patients were assigned to the high-risk group according to the INRG, and eight patients were assigned to the intermediate-risk group. No patient fell into the low-risk or very low-risk group (see Table 1).

A total of 40 tumor manifestations were evaluated in the 34 included patients. Ten cases were primary tumors, and 13 cases were local recurrences. In three cases, the tumor manifestation was lymph node metastases, and in 14 cases, there were distant metastases.

The mean volume of the tumor manifestations at their respective first examination time was 35.51 cm^3^ ± 74.05 cm^3^ (range: 0.67 cm^3^–321.27 cm^3^).

Of the 34 patients included in the study with 40 evaluable tumor manifestations, a total of 121 evaluable MRI examinations were available.

In total, 14 tumor manifestations were presented in only two examinations and could be evaluated. Moreover, 14 tumors were evaluated in exactly three examinations each. Two manifestations were presented together in 13 MRI examinations, which corresponds to the maximum number of evaluable examinations of a patient. The remaining 25 examinations were distributed among patients, with four to six examinations. Each tumor manifestation presented an average of 3.55 times (±2.40). On average, 1.16 ± 0.43 tumor manifestations were evaluated per MRI examination. This results from the fact that in 3 MRI examinations of one patient, three distinct lesions were evaluated, and in 13 MRI examinations of another patient, two lesions were evaluated. A total of three other patients also had two separate, distinct tumor manifestations and were also evaluated individually, but the two lesions were never displayed in the same MRI.

In total, 142 ADC values were analyzed from the 40 tumor manifestations (an average of 4.18 ADC values ±4.15 per patient; range: 2–26). Of these, 31 measurements were from prestaging examinations; 34 were from final staging examinations; 29 were from intermediate staging investigations; 41 were from follow-up investigations; and seven were from unspecified investigations.

The mean ADC of the tumors, regardless of the site of onset, of INSS stage IV patients (n = 27) was 147.71 × 10^−7^ mm^2^/s ± 60.95 × 10^−7^ mm^2^/s (min. 48.84 × 10^−7^ mm^2^/s; max. 383.33 × 10^−7^ mm^2^/s), and the mean ADC value of tumors from INSS stage III patients (n = 7) was 147.68 × 10^−7^ mm^2^/s ± 47.58 × 10^−7^ mm^2^/s (min. 76.88 × 10^−7^ mm^2^/s; max. 295.93 × 10^−7^ mm^2^/s). The difference was not significant (*p* = 0.6).

The mean ADC value of N-Myc-positive tumors (n = 11) was 138.94 × 10^−7^ mm^2^/s ± 57.37 × 10^−7^ mm^2^/s (min. 55.54 × 10^−7^ mm^2^/s; max. 279.68 × 10^−7^ mm^2^/s), and that of N-Myc-negative tumors (n = 17) was 151.91 × 10^−7^ mm^2^/s ± 61.73 × 10^−7^ mm^2^/s (min. 48.84 × 10^−7^ mm^2^/s; max. 383.33 × 10^−7^ mm^2^/s) (see Figure 2). Again, the difference was not significant (*p* = 0.2).

The mean ADC value of all measurements of tumors from high-risk patients according to INRG (n = 26) was 141.43 × 10^−7^ mm^2^/s ± 59.27 × 10^−7^ mm^2^/s (min. 48.84 × 10^−7^ mm^2^/s; max. 383.33 × 10^−7^ mm^2^/s) and that of tumors in intermediate-risk patients according to INRG (n = 8) was 169.26 × 10^−7^ mm^2^/s ± 50.86 × 10^−7^ mm^2^/s (min. 76.88 × 10^−7^ mm^2^/s; max. 295.93 × 10^−7^ mm^2^/s) (see Figure 2). This difference was significant (*p* = 0.02) (see Figure 3).

The mean values are listed in Table 2.

A statement on the course of the disease was available for 70 measured values, and a corresponding reference point (BL) for the calculation of *∆ADCvalues* could be determined.

Of these, 24 values were measured during or at the end of an AB therapy cycle, 17 during or at the end of a chemotherapy cycle, one during or at the end of a radio-chemotherapy cycle, two during or at the end of a MiBG therapy cycle, two after an autologous stem cell transplantation, and 25 in the context of follow-up examinations after the end of therapy.

Of the total of 70 *∆ADCvalues*, 5 were *∆ADCregress*, 50 were *∆ADCstable,* and 15 were *∆ADCprogress*; a total of 22 were *er∆ADCvalues*: 4 *er∆ADCregress*, 11 *er∆ADCstable*, and seven *er∆ADCprogress* (there was an average of 88.36 d ± 13.40 d between the reference point and the examination under consideration).

The mean difference of the ADC value measured in the first examination compared to the ADC value of the last examination of all tumor manifestations was 11.78 × 10^−7^ mm^2^/s ± 63.08 × 10^−7^ mm^2^/s, which corresponds to a mean increase of 20.43% ± 62.42%.

The mean difference of all ADC values compared with a BL, as described above (n = 70), was, on average, *∆ADC* = 3.36 × 10^−7^ mm^2^/s ± 40, 55 × 10^−7^ mm^2^/s. This corresponds to a mean increase of 3.86% ± 24.75%.

The mean value of the *∆ADC* values with regress (n = 5) was *∆ADCregress* = 48.45 × 10^−7^ mm^2^/s ± 40.64 × 10^−7^ mm^2^/s. This corresponds to a mean increase of 31.34% ± 22.45%.

The mean of the *∆ADC* values with stable disease (n = 50) averaged *∆ADCstable* = 8.65 × 10^−7^ mm^2^/s ± 26.70 × 10^−7^ mm^2^/s. This corresponds to a mean increase of 7.36% ± 20.62%.

The mean value of the *∆ADC* values with progress (n = 15) was *∆ADCprogress* = −29.31 × 10^−7^ mm^2^/s ± 53.92 × 10^−7^ mm^2^/s. This corresponds to an average drop of 16.96% ± 23.36%.

The mean values of the *∆ADC* values in their entirety and by course of the disease are listed in Table 3.

The differences between the mean *∆ADC* values (absolute and percentage) according to the course of the disease were significant between *∆ADCregress* and *∆ADCstable* (absolute *p* = 0.02 and percentage *p* = 0.01); between *∆ADCprogress* and *∆ADCstable* (absolute and percentage each *p* < 0.01); and between *∆ADCregress* and *∆ADCprogress* (absolute and percentage each *p* < 0.01) (see Figure 4 and Figure 5). The effect size was always at least moderately to predominantly large (r > 0.3).

The evaluation of the *er∆ADC* values analogously to the evaluation of the *∆ADC* values resulted in the values presented in Table 4.

The differences between the mean *er∆ADC* values (absolute and percentage) according to the course of the disease were significant between *er∆ADCregress* and *er∆ADCstable* (absolute and percentage *p* = 0.01) and between *er∆ADCregress* and *er∆ADCprogress* (absolute *p* < 0.01 and percentage *p* = 0.01). The effect size was always at least moderate to large (r > 0.3).

The differences between *er∆ADCprogress* and *er∆ADCstable* were not significant (absolute *p* = 0.11 and percentage *p* = 0.09) (see Figure 6 and Figure 7).

Furthermore, the *∆ADC* values were also examined, considering the course of the disease as described above, using the clinical and human genetic parameters that were collected. As can be seen in Figure 8, there was no significant difference in the ADC changes between these groups.

The *∆ADC* values were also examined according to the type of therapy while maintaining the order according to the course of the disease (Figure 9 and Table 5). For example, there was a mean increase in the ADC value of 57.32% in the case of both regression under antibody therapy and regression under chemotherapy, ±12.80% or 14.03% ± 7.31% versus a 23.41% ± 25.16% decrease in ADC in case of progression with antibody therapy, and by 25.88% ± 20.62% with chemotherapy. However, here too, no significant differences could be seen between the changes depending on the type of therapy.

## 4. Discussion

MRI is usually the method of choice as a staging examination for childhood tumor diseases, as well as for neuroblastoma. The reasons for this are the lack of radiation exposure compared to X-ray-based methods, the high soft-tissue contrast, and good availability, as well as reproducibility [10,31]. The protocols used often include diffusion-weighted sequences so that an ADC is available [13,17,32]. The diagnostic added value of the ADC maps varies greatly. A possible application with direct added value for the patient could be the monitoring of a neuroblastoma disease under therapy. For example, purely morphological methods often do not provide satisfactory information about treatment response or treatment failure in the first few months after the start of therapy or changeover. Here, the change in the mean ADC value can serve as an indicator, as has been shown in this study.

The results presented here were able to demonstrate a connection between the ADC changes and the response to therapy, even within the first 120 days (four months) after the start of therapy.

However, the results are limited by the strong variance of the measured ADC values. Indicatively, a high standard deviation, independent of the categorization, stands out among all ADC values (±63.08 × 10^−7^ mm^2^/s). There are multifactorial reasons for this: MRI examinations were obtained from multiple institutions and were created with devices from different manufacturers using different b values. Additionally, there were significant quality fluctuations and losses in terms of artefact avoidance and noise reduction. In particular, the performance of lengthy MRI sequences in children regularly suffers from compliance-related movement artifacts. The tumor inhomogeneity, in combination with the sometimes lower resolution of ADC maps, is a further source of strong fluctuations. Necrosis, cystic parts, and hemorrhages, as well as tumor desmoplasia, cannot always be reliably measured separately from solid tumor parts. Here, an attempt was made to ensure the most accurate possible delineation of the ROI by means of image fusion and subsequent control. Very small tumors could not be measured in isolated cases (see the exclusion criteria), so, for example, initial tumors in early stages and small, regressive residual tumors are possibly underrepresented. Furthermore, tumors in different localizations were examined. In addition to sometimes slightly different MRI sequences depending on the body, the local peritumoral tissue is naturally different. In the case of small tumors, it cannot be guaranteed that no local tissue is detected in the edge area of the ROIs. Diffuse and disseminated tumor infiltrates, especially in the case of metastases, are also possible. Here, tumor cells lie between local cells, so it can be assumed that the ADC value is not independent of the surrounding tissue.

In addition, classic limitations of the retrospective study design can also be identified: It was not possible to make a reliable chronological classification in relation to a therapy that took place for every examination, especially external ones. Missing or inaccurate information on therapies that sometimes took place in foreign-language countries can often no longer be completely and reliably reconstructed. Similar ambiguities about the long-term course of some tumor diseases could also not be resolved in some cases. In particular, follow-ups of other patients were often incomplete. Furthermore, molecular genetic investigations were not carried out for every patient, with the N-Myc status being assigned high prognostic relevance, which could consequently also be reflected in the ADC changes [6,7,9,33]. This led to a need for selective inclusion and exclusion criteria from the examinations and, thus, a smaller number of cases with analyzable measurements before and after therapy, despite an initially large collective. In particular, such examinations at the end of therapy with regression of the tumor were comparatively few. The poor prognosis and the expected low response to therapy of neuroblastomas in the advanced stages are also responsible for this.

Despite these limitations and obstacles, significant correlations between diffusion or ADC changes and response to therapy could be shown. A positive change in the ADC value corresponds to a higher probability of a therapy response and regression, and a drop in the ADC value corresponds to a greater probability of progression. In particular, there were no negative *∆ADC* values in the case of tumor regression. On the other hand, small changes in the ADC value or the value remaining the same were particularly common in stable diseases. These correlations were even significantly detectable after a maximum of 120 days after the start of therapy.

*∆ADCstable* was, on average, slightly positive, suggesting a minimal change to the ultrastructure of the tumor. This might be due to a minimal response to therapy which is too insignificant to warrant classification as regression according to INRC. However, this remains speculative.

Our findings are in line with the previous publications on this topic [22,23,24,25,26] and fit the basic theoretical considerations that result from histopathological findings and the physical basis of the technique.

The knowledge gained here suggests that the changes in the ADC value can be evaluated as an additional radiological marker of therapy response. In particular, significant correlations between ADC changes and response to therapy could also be seen in the first 120 days after the start of therapy. Classic image morphological criteria are often not sufficient enough during this period to make a reliable statement regarding a therapy response. Here, an ADC value determination could provide further information and, thus, help to give an early prognosis regarding the success of a given therapy. Such information is of the highest relevance for a potential change of therapy or its discontinuation.

Whether a reliable prognosis should be made based on the ADC changes alone and whether definitive consequences for the therapy can be derived from this must be examined on the basis of prospective studies. Progressive studies could shed light on whether there are significant benefits for the patient from a therapy change based on the ADC changes.

The theoretical, underlying histological and physical relationships could be identified, similar to other tumor entities, [21,22,24,34] by means of a comparative study between pathological findings (especially cell density), the ADC value, and therapy success before and after the start of therapy.

The development of artificial intelligence-supported systems using deep learning, which can currently be observed in the diagnostic subjects, also allows the correlations seen here to be further consolidated and expanded. Systems of this type can already accurately carry out a large number of volumetric tasks. The ADC value that can be obtained in this way over several layers promises to be less susceptible to artifacts. It is also possible to feed deep learning algorithms with a large amount of information, of which the ADC value of the tumor is merely one part. Correlations with histological, molecular pathological, laboratory, chemical, and image morphological criteria are possible. The predictive importance of the ADC value cannot yet be definitively foreseen, but research, such as the current study, is already promising in this regard:

## 5. Conclusions

We were able to demonstrate a clear connection between ADC changes and response to therapy in neuroblastoma, even within the first 120 days of therapy: A positive change in the ADC value corresponds to a higher probability of a therapy response and regression, and a drop in the ADC value corresponds to a greater probability of progression.

## Figures and Tables

**Figure 1 cancers-15-01940-f001:**
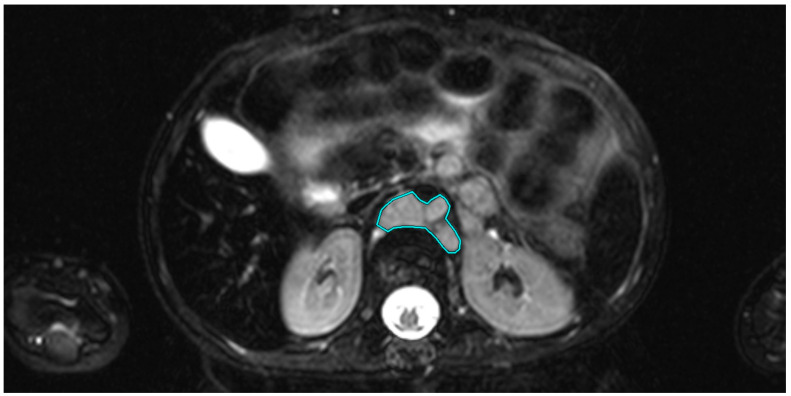
Axial T1-weighted image of a 2-year-old patient with stage IV neuroblastoma. The tumor is located para-aortally. The region of interest (ROI) is shown in turquoise, drawn using the “closed polygon” tool.

**Figure 2 cancers-15-01940-f002:**
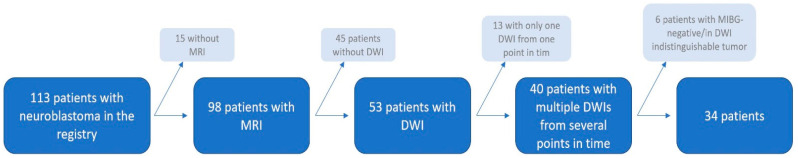
Flow chart illustrating the patient collective and the excluded patients.

**Figure 3 cancers-15-01940-f003:**
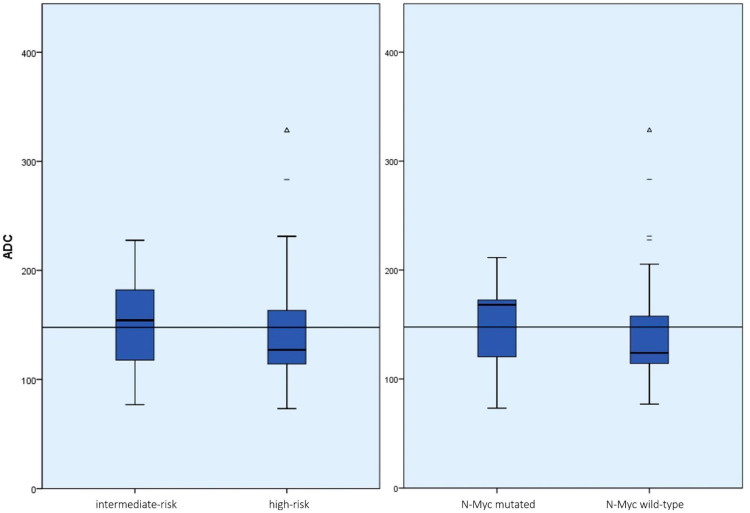
Boxplot representation of the ADC values of the tumor manifestation divided into INRG risk groups (**left**) and N-Myc status (**right**) with the mean of all measurements plotted as a reference. While the ADC values as a function of N-Myc status did not differ significantly (*p* = 0.2), the difference in ADC values for the INRG risk groups was statistically significant regardless of the location of the tumor (*p* = 0.02). (dashes = mild outliers; triangles = extreme outliers).

**Figure 4 cancers-15-01940-f004:**
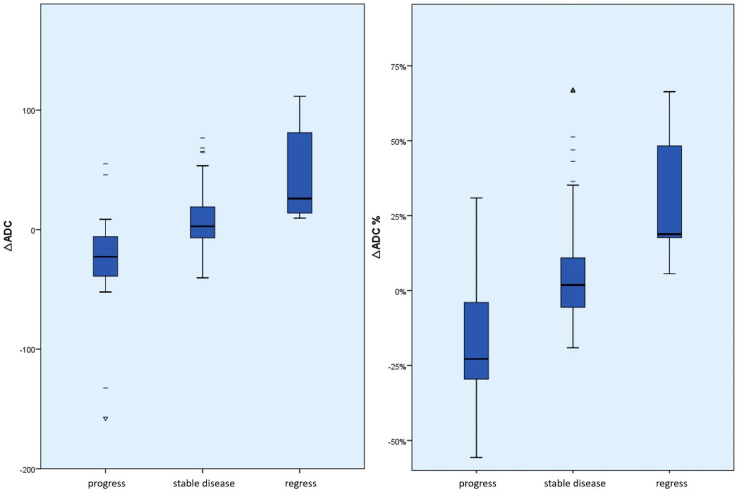
Box plot of the absolute or percentage *∆ADC* values according to the course of the disease. This shows an absolute and relative ADC increase in tumor regression compared to stable disease and tumor progression or an absolute and relative ADC decrease in tumor progression compared to stable disease and tumor regression. In particular, there were no negative *∆ADC* values in the case of tumor regression. (dashes = mild outliers; triangles = extreme outliers).

**Figure 5 cancers-15-01940-f005:**
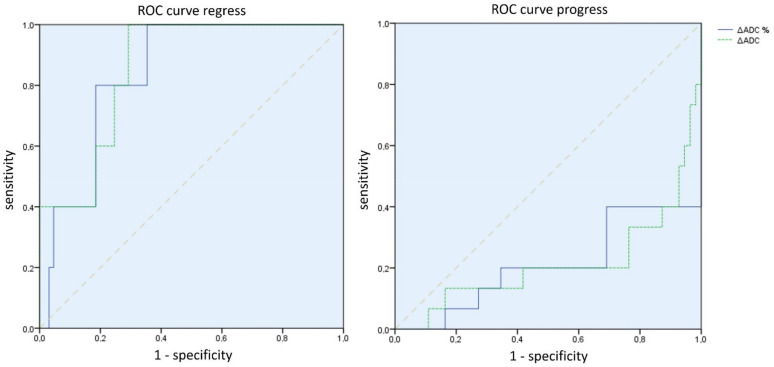
ROC curves of the absolute (green) and percentage (blue) *∆ADC* values for regress and progress prediction. It is shown that both an ADC decrease as a parameter in tumor progression and, in particular, an ADC increase as a parameter in tumor regression have a high ability to discriminate.

**Figure 6 cancers-15-01940-f006:**
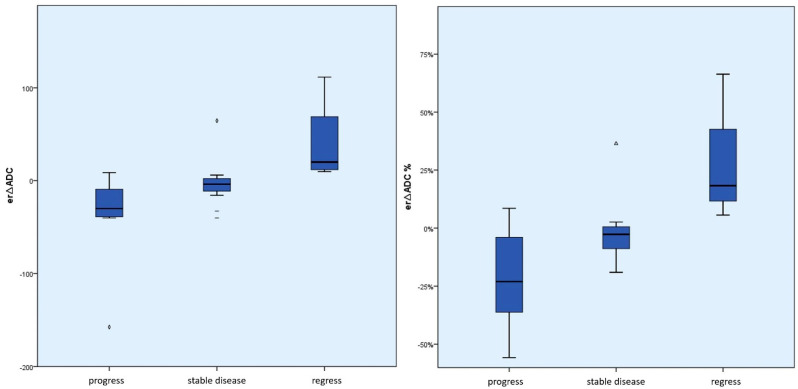
Box plot of the absolute or percentage *er∆ADC* values according to the course of the disease (outliers, min./max. in whiskers or individually). This shows an absolute and relative ADC increase in tumor regression compared to stable disease and tumor progression or an absolute and relative ADC decrease in tumor progression compared to stable disease and tumor regression. In particular, there were no negative *er∆ADC* values in the case of tumor regression. (dashes = mild outliers; triangles = extreme outliers).

**Figure 7 cancers-15-01940-f007:**
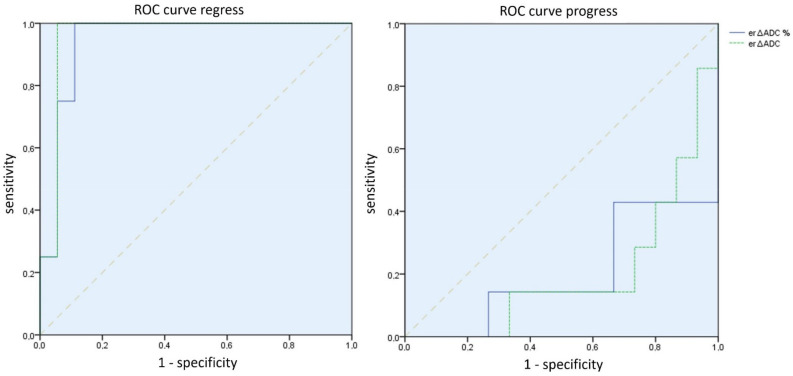
ROC curves of the absolute (green) and percentage (blue) *er∆ADC* values for regress and progress prediction. It is shown that both an ADC decrease as a parameter in tumor progression and, in particular, an ADC increase as a parameter in tumor regression have a high ability to discriminate.

**Figure 8 cancers-15-01940-f008:**
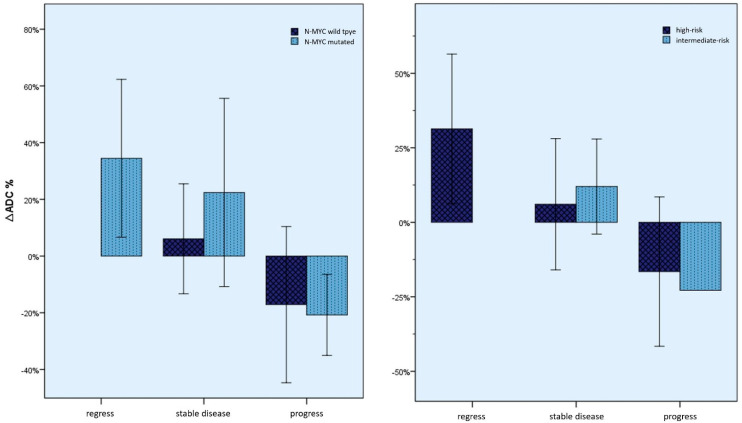
Mean percentage *∆ADC* values (whiskers = 1 SD) by N-Myc status and by INRG risk groups. Significant differences could not be seen here between the changes depending on the N-Myc status or the INRG risk group.

**Figure 9 cancers-15-01940-f009:**
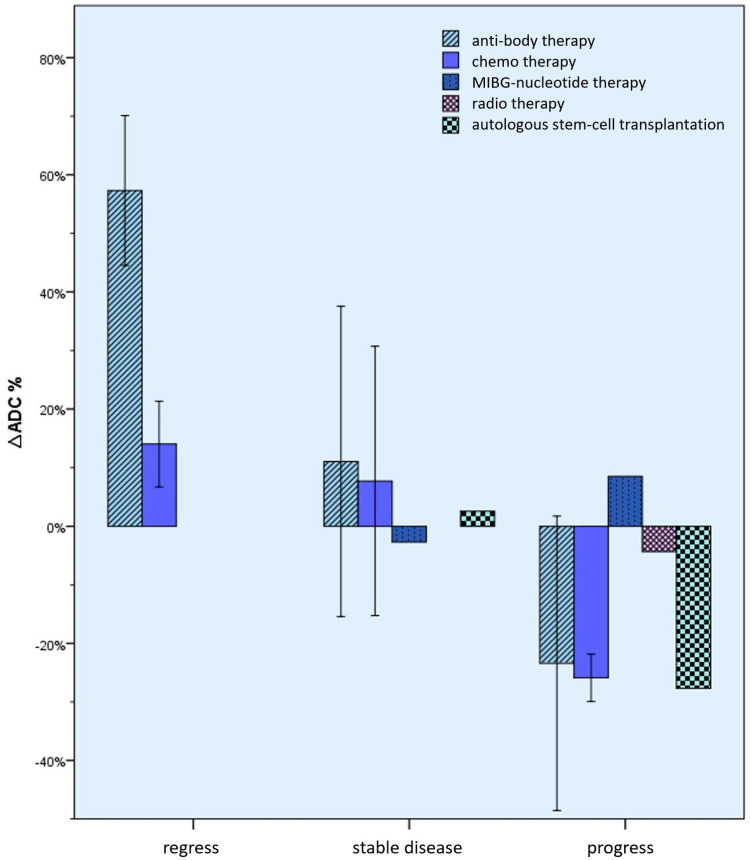
Mean percentage *∆ADC* values (whiskers = 1 SD) according to the type of therapy and disease progression. Here, too, significant differences could not be seen between the changes depending on the type of therapy.

**Table 1 cancers-15-01940-t001:** Distribution of patients by INSS, IRDFs, INRG, and N-Myc status.

Total	34
INSS Stage	
III	7
IV	27
IRDFs	
≤1	4
≥2	22
INRG Stage	
intermediate-risk	8
high-risk	26
N-Myc	
wild-type	17
mutated	11
NI	6

**Table 2 cancers-15-01940-t002:** Mean ADC values (bold) ± standard deviation (min.|max.) of all tumor manifestations depending on the clinical parameters and associated groupings or the molecular status. Within the respective parameter groups, there was only a significant difference in the ADC mean values stratified according to the INRG risk group between the high-risk group and the intermediate-risk group (*p* = 0.02).

Clinical Groups	n	Ø*ADC* ± SD [10^−7^ mm^2^/s]	Min.|Max. [10^−7^ mm^2^/s]	*p*
INSS stage				
4	27	**147.71** ± 60.95	48.84|383.33	
3	7	**147.68** ± 47.58	76.88|295.93	0.6
N-MYC				
positive	11	**138.94** ± 57.37	55.54|279.68	
negative	17	**151.91** ± 61.73	48.84|383.33	0.2
INRG risk group				
high-risk	26	**141.43** ± 59.27	48.84|383.33	0.02
intermediate-risk	8	**169.26** ± 50.86	76.88|295.93	

**Table 3 cancers-15-01940-t003:** Mean *∆ADC* values (bold) in absolute and percentage terms based on the course of the disease. There was a significant ADC increase in tumor regression compared to stable disease and compared to progress (absolute *p* = 0.02 or <0.01, percentage *p* = 0.01 or <0.01).

	n	Ø*∆ADC* ± SD [10^−7^ mm^2^/s]	Ø*∆ADC* ± SD [%]	*p* Compared to Regress (Absolute|Percentage)
total	70	**3.36** ± 40.55	**3.86** ± 24.75	
regress	5	**48.45** ± 40.64	**31.34** ± 22.45	
stable disease	50	**8.65** ± 26.70	**7.36** ± 20.62	0.02|0.01
progress	15	**−29.31** ± 53.92	**−16.96** ± 23.36	<0.01|<0.01

**Table 4 cancers-15-01940-t004:** Mean *er∆ADC* values (bold) in absolute and percentage terms based on the course of the disease. There was a significant absolute and relative ADC increase in the first 120 days after the start of therapy with tumor regression compared to stable disease and compared to progress (absolute *p* = 0.01 or <0.01, percentage *p* = 0.01).

	n	Ø*er∆ADC* ± SD [10^−7^ mm^2^/s]	Ø*er∆ADC* ± SD [%]	*p* Compared to Regress (Absolute Percentage)
total	22	**−6.78** ± 48.21	**2.99** ± 25.44	
regress	4	**40.28** ± 48.03	**27.11** ± 26.84	
stable disease	11	**−3.11** ± 26.85	**−2.14** ± 14.72	0.01|0.01
progress	7	**−39.44** ± 55.10	**−21.51** ± 23.21	<0.01|0.01

**Table 5 cancers-15-01940-t005:** Mean *∆ADC* values (bold) ± SD according to the type of therapy and the course of the disease. Significant differences could not be seen between the changes depending on the type of therapy.

Ø*∆ADC* ± SD	Anti-Body Therapy	Chemo Therapy	MIBG-Nucleotide Therapy	Radio Therapy	Stem-Cell Transplantation
n	23	17	2	1	2
total [10^−7^ mm^2^/s]	**−0.20** ± 62.48	**2.29** ± 20.78	**2.36** ± 8.86	**−5.73**	**−13.68** ± 27.76
total [%]	**3.09** ± 33.94	**4.89** ± 22.61	**2.92** ± 7.94	**−4.31**	**−12.55** ± 21.40
regress [10^−7^ mm^2^/s]	**96.35** ± 21.52	**16.52** ± 8.54			
regress [%]	**57.32** ± 12.80	**14.03** ± 7.31			
stable disease [10^−7^ mm^2^/s]	**12.85** ± 34.62	**4.77** ± 17.64	**−3.91**		**5.95**
stable disease [%]	**11.06** ± 26.50	**7.73** ± 22.99	**−2.70**		**2.57**
progress [10^−7^ mm^2^/s]	**−45.55** ± 69.41	**33.96** ± 5.32	**8.63**	**−5.73**	**−33.31**
progress [%]	**−23.41** ± 25.16	**−25.88** ± 4.05	**8.54**	**−4.31**	**−27.68**

## Data Availability

Data is unavailable for sharing due to privacy and ethical restrictions.

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
