# Peer review of "Quantitative Diffusion-Weighted MRI of Neuroblastoma"

_cancers, 2023, doi:10.3390/cancers15071940_

Round 1

Reviewer 1 Report

Interesting and well written study reporting a large experience with a relatively rare tumor.

The study is well and deserves interest to the readers. The unique comment is about the abstract that is a bit too long. 

Author Response

Thank you for reviewing our manuscript.

The abstract was significantly shortend.

Reviewer 2 Report

The manuscript “Quantitative Diffusion-Weighted MRI of Neuroblastoma” reports an interesting study on early change in the ADC values in neuroblastoma under therapy. The study reports a systematic retrospective evaluation of treated neuroblastoma cases grouped as pre-stage, intermediate stage, final stage, and follow-up stage. The study includes 34 patients with 40 evaluable tumor manifestations and diffusion-weighted MRI examinations were finally included. Authors concluded that a significant increase in the ADC value in the case of regression and a significant decrease in the case of progress.

1.     Line 261-263, I believe that average age 6.8± 5.8 years is quite confusing. I would rephrase in range like 1-26 years old.

2.     Fig. 4 &5, Statement “there were no negative ΔADC values in the case of tumor regression”. This is quite exciting result, however, the ΔADC increases in stable disease condition, why?    

3.     Discussion line 430-433; statement “Despite some limitations, the results presented here were able to demonstrate a clear connection between the ADC changes and the response to therapy, even within the first three months after the start of therapy.”  Result does not support the conclusion of significant change in ADC within 3 months of therapy. I wonder if authors can show the results based on their categorization of grouped as pre-stage, intermediate stage, final stage, and follow-up stage, and discuss the results.

4.     Discussion should be based on findings of results. It seems like authors elaborated just limitations of the study in discussion section.

5.     Line 435-454, Authors agree that “the results are limited by the very strong variance of the measured ADC values…..” Then who confidently a physician could readout therapeutic outcome based on change in ACD during therapy.     

Reviewer 3 Report

The title represents the manuscript contents, indicating the study design and its innovative results.

The abstract is accurate and concise, providing an informative and balanced summary of the study.

In the introduction the authors describe the scientific background and clearly explain the scientific hypotheses.

Clear and exhaustive exposition of materials and methods, inclusion and exclusion criteria.

Rigorous analysis of MRI images using specific software.

Careful statistical analysis.

Clear presentation of results. They are presented in a transparent and unbiased manner.

The discussion summarizes the key results and its relevance.

The results are explained point by point with reference to similar articles recently published on the subject.

The conclusions are warranted by the results.

The references are properly cited.

The images, diagram, tables and graphs are done well according to the editorial standards of the magazine.

Author Response

Thank you for reviewing our manusript and providing point to point feedback.

Some changes were made to the abstract and the discussion part.

Round 2

Reviewer 2 Report

This revised version of the manuscript responses the reviewer's comments satisfactorily, therefore recommended for acceptance for publication of the article.